# Single V2 defect in 4H silicon carbide Schottky diode at low temperature

Timo Steidl[1,10], Pierre Kuna [1,10], Erik Hesselmeier-Hüttmann[1], Di Liu [1,2], Rainer Stöhr[1], Wolfgang Knolle [3], Misagh Ghezellou [4], Jawad Ul-Hassan [4], Maximilian Schober [5], Michel Bockstedte [5], Guodong Bian[6,7,8], Adam Gali [6,7,8], Vadim Vorobyov [1] ✉ & Jörg Wrachtrup[1,9]

Nanoelectrical and photonic integration of quantum optical components is crucial for scalable solid-state quantum technologies. Silicon carbide stands out as a material with mature quantum defects and a wide variety of applications in semiconductor industry. Here, we study the behaviour of single silicon vacancy (V2) colour centres in a metal-semiconductor (Au/Ti/4H-SiC) epitaxial wafer device, operating in a Schottky diode configuration. We explore the depletion of free carriers in the vicinity of the defect, as well as electrical tuning of the defect optical transition lines. By detecting single charge traps, we investigate their impact on V2 optical line width. Additionally, we investigate the charge-photon-dynamics of the V2 centre and find its dominating photon-ionisation processes characteristic rate and wavelength dependence. Finally, we probe the spin coherence properties of the V2 system in the junction and demonstrate several key protocols for quantum network applications. Our work shows the first demonstration of low temperature integration of a Schottky device with optical microstructures for quantum applications and paves the way towards fundamentally scalable and reproducible optical spin defect centres in solids.

Over the recent decades, colour centres have matured into a valuable resource for quantum applications[1]. However, hosted in solids they are subject to the fluctuating electron charge environment[2,3] which deteriorates their quantum properties. Elimination of charge fluctuations in the host material is key for the fundamental scalability of the platform. Silicon carbide (SiC) is a mature and well-developed semiconductor material system[4]. Due to its unique combination of optical, semiconductor and material properties, it is an excellent host for quantum emitters and spin defects[5]. Importantly, the ability to control the Fermi level, i.e., the charge environment of defects by controlled doping is advantageous for integrating optically addressable solid

state defects. For example, in-plane Schottky junctions in diamond, in conjunction with surface termination were shown to be effective for defect charge state switching[6]. In SiC, integration of defect ensembles with PIN junctions was reported[7], leading to electrically detected magnetic resonance, and precise charge control. Recently, p⁺-p-n⁺ structures[8] were tested for integration with G centres in silicon on an insulator material platform. For quantum optical applications the optical emission linewidth is a key parameter. Spectrally narrow and stable emission lines enable e.g. large entanglement rates in quantum repeater networks. Previously, PIN junctions were studied at temperatures of a few Kelvin in commercially available SiC samples, for

---

[1]3rd Institute of Physics, IQST, and Research Center SCoPE, University of Stuttgart, Stuttgart, Germany. [2]John A. Paulson School of Engineering and Applied Sciences, Harvard University, Cambridge, MA, USA. [3]Department of Sensoric Surfaces and Functional Interfaces, Leibniz-Institute of Surface Engineering (IOM), Leipzig, Germany. [4]Department of Physics, Chemistry and Biology, Linköping University, Linköping, Sweden. [5]Institute for Theoretical Physics, Johannes Kepler University Linz, Linz, Austria. [6]HUN-REN Wigner Research Centre, Budapest, Hungary. [7]Institute of Physics, Department of Atomic Physics, Budapest University of Technology and Economics, Budapest, Hungary. [8]MTA-WFK Lendület -Momentum- Semiconductor Nanostructres Research Group, Múzeum körút, Hungary. [9]Max Planck Institute of Solid State Research, Stuttgart, Germany. [10]These authors contributed equally: Timo Steidl, Pierre Kuna. ✉e-mail: v.vorobyov@pi3.uni-stuttgart.de

precise charge control of di-vacancy (VV⁰) centres[9]. Usually, the application of a negative bias to deplete charges in the intrinsic region of the PIN structure leads to the creation of defects in an ionised charge state which is not emitting photons[10]. While the silicon vacancy (V2) defect in 4H-SiC is a promising scalable spin photon interface platform[11,12] with coherent optical transitions up to $T = 20$ K[13], single V2 centres have not been investigated in semiconductor junctions.[7,14]

In this work, we investigate the operation of single silicon vacancy V2 centres in a Schottky diode integrated with an optical micro solid immersion lens at low temperature. We observe that charge depletion yields narrowing of the optical transition lines and furthermore investigate the electrical tuning of the defect via the DC Stark shift. By localising the defect optically we precisely map the depletion zone around the Schottky contact and calibrate its local intrinsic doping concentration. We find that the defect stays in its negative charge state throughout the range of voltages used in the investigation. We support our findings with a theoretical and experimental study of photo-ionisation dynamics of single defects at various excitation wavelengths. Finally, we show the integration of the Schottky junction with an optical microstructure and demonstrate its use in typical experimental protocols relevant for the operation of a quantum network, including the measurement-based optical transition stabilisation, electron spin coherent manipulation and nuclear spin repetitive single-shot readout. With this, we combine optical, electrical, electron and nuclear spin control of isolated V2 defects in silicon carbide.

## Results

The V2 centre is a silicon vacancy ($V_{Si}$) at the cubic lattice site of 4H-SiC epitaxially grown layer. It is a very good single photon emitter[15] with long spin memory lifetimes[12]. In our study, we investigate V2 next to a Schottky contact realised by a Au-Ti-SiC interface. The electric circuit is closed by semi-insulating silicon carbide (schematically shown in Fig. 1a). At cryogenic temperatures, the defect states can be excited efficiently by resonant optical excitation at 1.352eV (~916 nm) with spin-resolved $A_{1,2}$ optical transitions for $m_s = \pm 1/2, \pm 3/2$ spin manifolds (Fig. 1b) split by ~1 GHz[13]. We scan both transitions simultaneously with two resonant lasers separated by excited state zero field splitting yielding a single resonance peak in the photoluminescence excitation (PLE) spectrum shown in Fig. 1c. Exploring the current-voltage (I-V) characteristic of the diode (Fig. 1d). At low temperatures the diode threshold forward voltage increases compared to room temperature, matching well with the diode law. If too much voltage is applied and

the current exceeds >mA, heating and electroluminescence can be observed (compare Supplementary Fig. 5). Using confocal microscopy (Fig. 1e) we localise single defects close to the metal contact. We perform subsequent PLE measurements at different bias voltages. The extracted PLE positions and linewidths are shown in Fig. 2a for two defects with defect-1 located ~11.2 μm from the contact (red circle in Fig. 2b) and defect-2 with a distance of ~4 μm to the electrode (see Supplementary Fig. 1b). We apply a total voltage sweep from −150 V to +30 V. For negative bias, the detuning follows a purely quadratic dependence, with a Stark coefficient (polarisability) which is extracted to be $-0.047 \pm 0.006$ GHz/$\left(\frac{MV}{m}\right)^2$. The local effective electric field strength has been determined using a COMSOL[16] simulation. As the electric field components point perpendicular to the crystal c-axis this is the first time a perpendicular Stark tuning coefficient for V2 centres in 4H-SiC is reported, with a linear parallel Stark coefficient of 3.7 GHz/$\left(\frac{MV}{m}\right)$ reported in[17]. At near-zero or forward bias, a PLE spectral broadening is observed. Whereas at negative bias, we typically observe a line narrowing from 170 MHz down to 40 MHz. The linewidth of defect-1 is measured to be 40 MHz, slightly higher than the Fourier limit (~20 MHz). The effective lifetime limit of the single PLE curve will lie in between the values of each single optical transition (blue line in Fig. 2a, for calculation see Supplementary Note 1). The most narrow optical line (purple squares in Fig. 2a) is observed at defect-2 and reaches the $A_2$ Fourier limit of around 14 MHz.

The line narrowing effect is reproducible with almost no hysteresis (see Supplementary Fig. 2a). PLE narrowing can be explained by the depletion of the surrounding volume charge in the vicinity of the defect. The depletion zone size around the Schottky contact rises with larger negative bias voltage (see Supplementary Fig. 2e). For the specific defects investigated, the zone in which line narrowing occurs reaches its position at around $U_{d_1} = -10$ V ($U_{d_2} = -4$ V), which we denote as depletion voltage. However, for the other 30 investigated defects across the sample, the depletion voltage varies. We combine all data in Fig 2b, where we depict the PLE spectral linewidth depending on the applied voltage and distance from the contact. The experimentally measured linewidth shows a threshold-like behaviour, at voltages predicted by simulations (green dashed line, see Supplementary Fig. 2c-d), once the edge of the depletion zone reaches the defect location, lines get narrow for voltages below depletion voltage. From comparing simulation and our experimental results we confirm the estimated doping concentration of the intrinsic layer to be $[N_{Epi}] = 7 \times 10^{13}$ cm$^{-3}$. The mechanism for line narrowing is best exemplified by

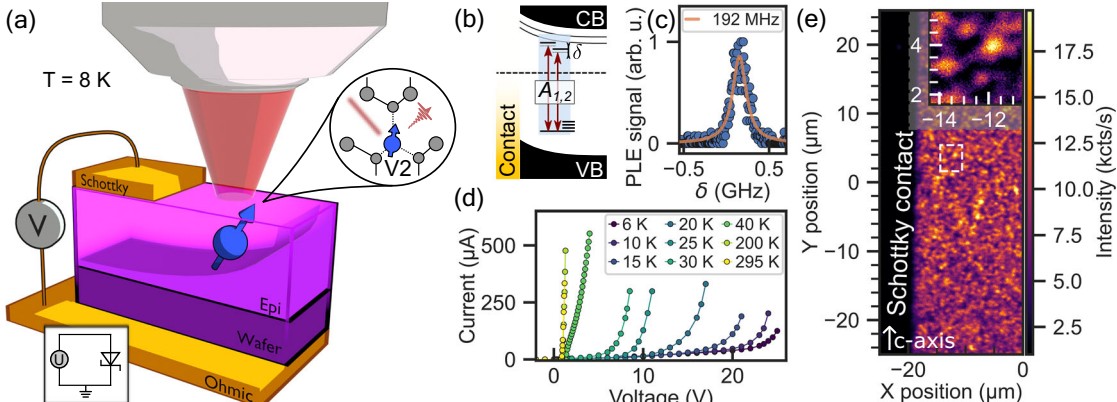

**Fig. 1 | V2 colour centre in Schottky diode. a** Schematic view of V2 colour centre with depletion zone of charge carriers formed next to the barrier and simple circuit diagram. **b** Schematic band gap bending diagram in the vicinity of the junction with mid gap defect states. Allowed transitions $A_{1,2}$ with detuning $\delta$. **c** Photoluminescence excitation spectrum (PLE) of single V2 centre with $A_1$ and $A_2$ simultaneously scanned, linewidth extracted using Lorentzian fit. **d** Current-

voltage (I-V) characteristic of the obtained Schottky diode, showing the forward biased current, measured at several temperatures from room temperature down to cryogenic environment. **e** Confocal map of the studied sample A in the vicinity of the Schottky contact. Inset: zoomed view on marked (white dashed rectangular) to identify defect locations.

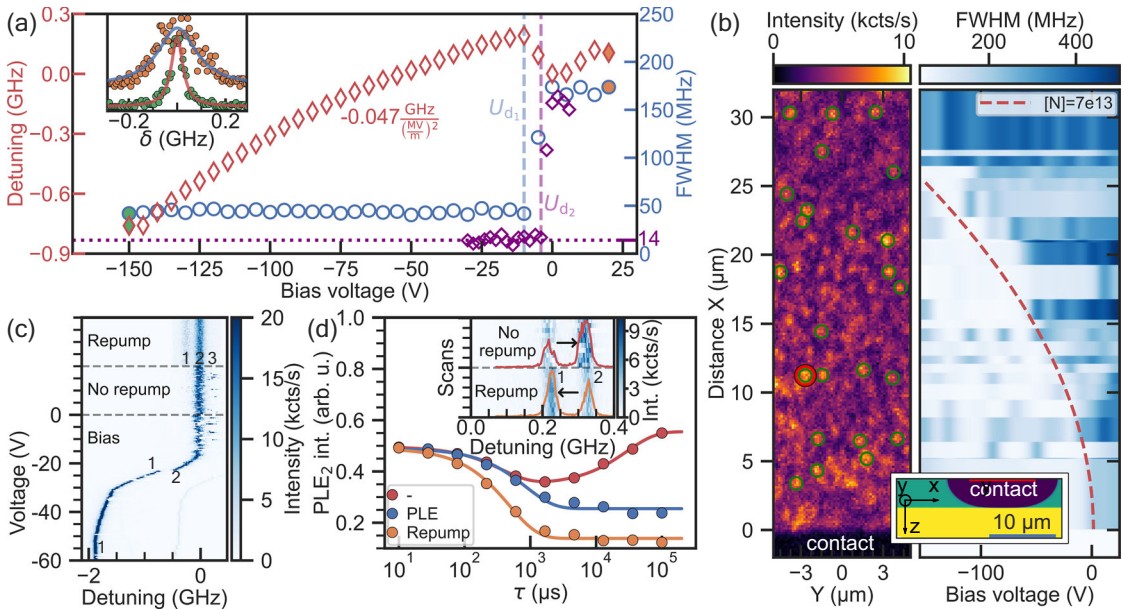

**Fig. 2 | Optical spectroscopy of V2 centre in a Schottky diode. a** Extracted linewidth and spectral PLE position of defect-1 marked with a red circle in (**b**) versus applied voltage ($-150$ V to $+20$ V). Detuning follows a purely quadratic dependence yielding a perpendicular Stark tuning coefficient of $-0.047 \pm 0.006$ GHz/$\left(\frac{MV}{m}\right)^2$. Inset show both examples, broad PLE without bias and narrow PLE lines in the depleted case (voltage data point indicated with filled markers). Blue octagons show measurement of defect-2, which has a lifetime limited linewidth (calculation uses lifetimes from ref. 27, see Supplementary Note 1 in Supplementary information) (**b**) Right: boundary of the depletion zone mapped by single defects (shown on the left) and simulated with COMSOL[16] (red dashed line is the threshold for $[e^-] = 1 \times 10^{12}$ cm$^{-3}$). The COMSOL simulation has been parametrically optimised, with best agreement for 2 μm deep colour centres and initial epi-layer doping concentration (green colour of inset) of $[N] = 7 \times 10^{13}$ cm$^{-3}$, validating given sample properties. Substrate doping (yellow colour) is $[N] = 1 \times 10^{17}$ cm$^{-3}$. **c** Exemplary PLE measurements of defect from sample B with distinct splitting of the resonance intro three

equally spaced lines (1,2,3), caused by a nearby charge trap, showing occasional charge trap ionisation and jumps between lines 2-3. Application of off-resonant (repump) laser results in motional averaging and a steady state charge distribution between lines 1-2, whereas negative bias applied to the diode stabilises the charge state of the trap and eventually narrows down the PLE line. **d** Inset: PLE of a V2 centre affected by a fast-switching two-level system (TLS) in its vicinity, again associated with a charge trap. Solid curves show the sum of all lines and highlight the change of the PLE priority with and without weak CW repumping. Mainfigure: time-resolved study of that TLS by tracking the second PLE intensity versus the duration of laser illumination (red: no excitation; orange: 20 μW off-resonant excitation; blue: 3 nW of resonant to V2 excitation). Depopulation of the corresponding TLS state (decrease in PLE intensity) by photo-induced reconfiguration, while recovery if no laser is applied. Single exponential fits have been used to plot the solid lines.

single defects with PLE spectral lines further split by strongly coupled nearby electron charge traps (see Fig. 2c). The figure shows a PLE spectrum of a single defect with three PLE lines. We confirm that each line belongs to the same defect, by performing hyperfine-resolved optical detected magnetic resonance (ODMR), see Fig. S4h. We attribute this behaviour to the presence of a single nearby fast switching charge trap, producing a discrete Stark shift of the defect[18,19]. As we see three PLE positions (Fig. 2c lines 1–3) we assume the charge trap to be a carbon vacancy $V_C$ as it has three stable charge states $(2+, +, 0)$[20]. The charge trap dynamics can be accelerated by laser illumination[21] leading to motional averaging and redistribution of its steady state population distribution[3]. For zero bias voltage and no additional laser, we observe that the V2 PLE is mostly found in the centre position (line 2), showing only rare shifts to other spectral positions (line 3). Applying a weak repumping laser during the PLE scan partly populates the charge trap into the third charge state (line 1). The population in line 3 can not be observed under this condition. As soon as the depletion zone reaches the defect (causing a jump in the PLE and a linewidth narrowing) the charge trap is depleted so that just one line remains visible. For another V2 defect, two PLE positions can be observed at zero bias (inset of Fig. 2d). In the following, we explore the temporal dynamics of the trap charge states. We applied the following sequence: a charge resonance check (CRC1) at the right line, then repumping, applying resonant laser or just waiting for a certain time and finally another resonance check at the right line (CRC2). One can observe a depopulation if using a laser pulse and a recovery if no laser is applied. The timescale of charge transfer is on the order of a few milliseconds. While

the traps switch the charge state within the depletion zone the V2 defect does not change its charge state dynamics at the applied voltage bias (see Fig. S3a). This is different for other defects integrated into semiconductor devices[8,10].

To further gain insight, we perform ionisation rate measurements under various wavelengths and laser powers. Figure 3a shows the power-dependent ionisation rate under resonant excitation only. Measurements have been taken at several bias voltages showing no significant change, indicating that the underlying processes are independent of the defects charge environment. Figure 3b shows the ionisation rate for a case where we apply an additional laser pulse together with the resonant laser. Using two adjustable single-mode lasers (Toptica CTL, Hubner C-Wave) we were able to use different wavelengths between 880 nm and 1200 nm for redistribution of charges in the vicinity of the defect (see Fig.3). Each point marks the slope of a power-dependent ionisation rate. There is a significant drop for wavelengths beyond 950 nm which corresponds to a threshold energy for ionisation of 1.31 eV. This perfectly matches the ab-initio calculations for the ionisation cross-section from the negative charge to the double negatively charged state $(-) \rightarrow (2-)$ shown in Fig. 3d. Discrepancies between the measured ionisation rate and calculated values $(\Gamma_{PI})$ might occur due to power calibration (measured outside the cryo versus expected intensity at the defect) or no perfect occupation of the excited state and competing processes like repumping. While ionisation to the neutral state is also possible, theory predicts a tenfold smaller rate compared to our experimental observation. Additionally, the defect would transform from the $V^0$ state

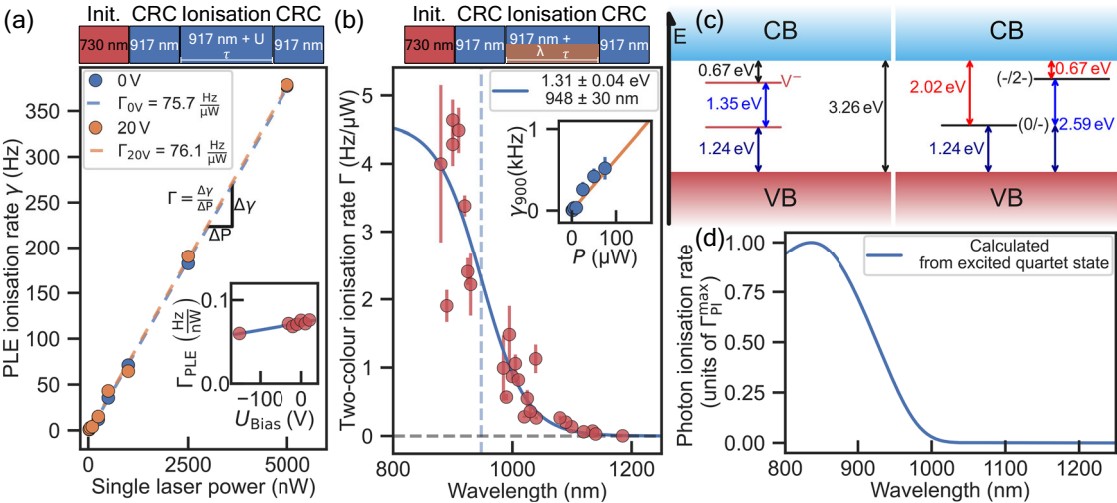

**Fig. 3 | Ionisation behaviour of single V2 centre under PLE and two-colour laser irradiation. a** Ionisation rate $\gamma$ under resonant laser irradiation versus PLE laser power shows almost identical results for two different bias voltages (inset shows line slope $\Gamma$ for various voltages and a decreasing trendline (blue) for lower voltages). $\gamma$ is determined by sweeping the ionisation pulse length $\tau$ and extracted using an exponential fit. **b** Two colour absorption power-dependent ionisation rate $\Gamma$ (slope of $\gamma_\lambda$) versus wavelength (error bars $\hat{=} \pm 1\sigma$). Second laser pulse (varied in length, wavelength and power) is applied together with PLE lasers at low power. Fit with Fermi distribution yields a threshold of 948 nm for the charge conversion of $V_{Si}^-$ to $V_{Si}^{2-}$, compare (**c**). Schematic pulse sequences for (**a**, **b**) shown on top. Inset: exemplary ionisation rate $\gamma$ versus power of second laser for a wavelength of $\lambda = 900$ nm again linearly fitted. All error bars are equivalent to $\pm 1\sigma$. **c** Defect level and charge state transition level picture of the cubic silicon vacancy centre within the band gap (3.26 eV used). Values for the (−/2−)-transition taken from ref. [7], the other values are deduced from that. **d** Ab-initio calculation of the photon absorption cross-section for the charge transition (−/2−) of V2, recalculated to a photo-ionisation rate $\Gamma_{PI}$ (more detailed information in Supplementary Note 4). Similar wavelength dependence is visible like for measured data in (**b**).

immediately back to the bright charge state under resonant excitation at 1.352 eV. From this, we can infer that the defect predominantly ionises into the dark state $V^{2-}$. Depletion of the electron density in the vicinity of the defect thus further reduces the probability and prevents the defect from being in the dark state, evident from reduced ionisation rate (inset of Fig. 3a) and confocal maps at negative bias (see Supplementary Fig. 3).

Next, we demonstrate the integration of the Schottky diode and a microscopic solid immersion lens (SIL). The same sample has recently been used to show nuclear assisted single-shot readout[11,22]. The main motivation here was to confirm that electrical and optical structures can be combined (Fig. 4a) leading to a tuneable spin photon interface platform based on V2 centres in SiC. The challenge here is to ensure that the defect is located in the depletion zone, while not deteriorating the quality of the optical structure by metallic contacts, and ensuring the excellent spin coherence properties of the defect (see Supplementary Note 1). Similar to previously reported, the defect electron (nuclear) spin, shows 0.4 ms (70 ms) of Hahn echo $T_2$ time, which can be further extended via the dynamical decoupling[23] (see Fig. 4b and Supplementary Fig. 4) limited by magnetic noise from a bath with natural abundance of nuclear spin isotopes. By applying the negative bias, we confirm that those parameters are preserved within the depletion zone of the junction, crucial for applications.

## Discussion

Our work gains insight into the charge state dynamics and stability of the silicon vacancy. This forms a clearer picture of the system, which was only speculated before. Additionally, the present device structure allows to reconstruct previously reported effects (e.g., Stark tuning, linewidth narrowing,…) using a much simpler manufacturing process. It allows chip-industry-like scalability and interoperability with required photonic technologies, which is an additional benefit of using a Schottky type of junction.

The effect of the optical line narrowing has its most implications on optical readout and emitted coherent photons quality. Its importance is evident from the clear improvement in the resonant charge

state readout (CRC) and nuclear spin repetitive readout (SSR) photon statistics presented in Fig. 4c–d. For the charge state readout, the raw photon statistics accumulated over several hours of measurement time show a well-defined Poissonian lobe for the case of U = −30 V of bias voltage, while showing a decaying distribution for unbiased cases, caused by significant spectral diffusion. For the readout of the nuclear spin state, due to the better stabilised optical line position, during the lengthy measurements, the figure demonstrates a brighter and better defined photon histogram with a drastic change in the middle overlapping region for different quantum states Fig. 4d.

It is crucial for spin measurements and spin photon entanglement protocols, that optical transitions would not spectrally shift (diffuse) during the whole measurement run. Alternatively, the excitation and emission rates of defects would drift causing an infidelity in calibrated parameters. Furthermore, with a spin photon entangled state, the spectral diffusion leads to unwanted phase accumulation on the photons, leading the dephasing of the entangled state. Normally, for solid state defects, it is ensured by applying a charge-resonance check measurement, which probes the fluorescence level. When the number of photons exceeds a certain threshold, defect resonance is considered to be within a small interval centred around the laser frequency with a high degree of confidence. As a typical example, here we find the success rate of finding a defect resonance within its natural halfwidth interval around the laser frequency and find that with confidence of 99.85% (above $\mu + 3\sigma$) at the depleted case 60.4% of the entanglement attempts would be successful while for the default case it is only 35 %. Figure 4e bottom panel. This could be even more important for samples with larger intrinsic nitrogen concentration and is affecting the entanglement link efficiency directly, as less entanglement attempt per channel is required for a success.

We attribute the beneficial charge robustness of the silicon vacancy to its favourable environment. The available charge traps, namely carbon vacancy, in its vicinity support the $V_{Si}$ to be most of the time in the correct charge state. Investigations of the probability of ionisation of the carbon vacancy (see Fig. S7) reveal the defect to be a donor-like neighbour under photo illumination, further exclude $V_{Si}^0$.

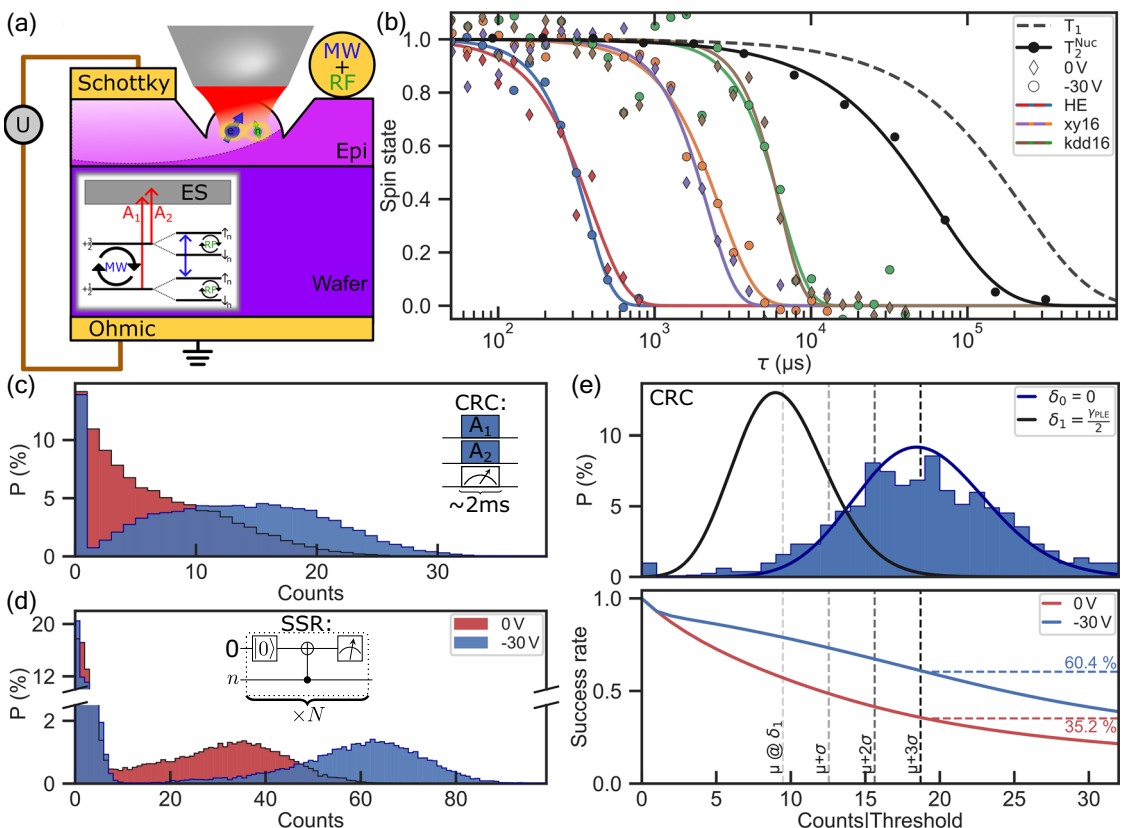

**Fig. 4 | Combination of electrical and optical structures and observation of spin properties under negative bias. a** Schematic drawing of a sample showing a colour centre integrated into a SIL within the depletion zone of the Schottky contact. Inset: ground state level structure with spin hyperfine splitting. **b** Electron and nuclear spin memory coherence measured with various dynamical decoupling sequences. The dashed line shows the $T_1$ limit. All data sets are fitted by stretched exponential decays. **c** Charge resonant check histogram for 0 V and -30 V accumulated over several hours, showing higher count rates for the depleted case. **d** Single-shot readout (quantum circuit diagram in inset) histogram for both voltages. Using the more distinct Poissonian distribution for negative bias allows

higher measurement rates by more favourable pre-selecting thresholds. **e** Top: blue histogram shows best CRC measurement which can be attributed to the best frequency overlap of resonant laser and optical transition. Simulation of CRC measurement with a laser detuned by the defect's natural halfwidth half-maximum (HWHM). Bottom: success rate is defined as the fraction of the total SSR events which will be used for data processing, as a function of the chosen collection threshold for normal and depleted case. Dashed lines show first and second moment of photon counts distribution of defect detuned by half width half maximum. Depleted colour centres yield a measurement fidelity of more than 99% ($\mu + 3\sigma$) without losing more than half of SSR events.

The charge transition behaviour between these three states, fits the observed results of PLE distribution (see Fig. 2c) for a V2 affected by the emerging electric field discrepancies.

In conclusion, our findings mark novel insights into the charge state dynamics of the V2 centre, underlining the application relevant charge state stability[5]. These observations make combinations of V2 centres with electronic structures in SiC a promising system for quantum network applications and spin photon interface[24]. The observation of discrete charge traps opens a new possibility for the optical addressability of dark colour centres, potentially suitable for forming a larger spin cluster around the V2 system[18,19]. Our results pave the way for the integration of junctions with nano beams which promise high collection efficiencies[25,26] (see Supplementary Fig. 1) and unleash the full potential of the material platform.

## Methods

### Experimental setup

All experiments were performed at cryogenic temperature <10 K (if not stated differently) in a Montana Instruments cryostation. A self-build confocal microscope was used to optically excite single V2 centres and detect the red-shifted phonon side band. Initialisation of the charge state is performed by off-resonant excitation via a 728 nm diode laser (Toptica iBeam Smart). For resonant optical excitation we used an external cavity tuneable diode laser (Toptica DL Pro), which was split

and frequency shifted by two separate AOMs to address both optical transitions selectively. Those transitions are called $A_1$ and $A_2$ and are split by $\approx 1$ GHz, depending on the zero-field splitting of the excited state. Laser photons are filtered by two tunable long-pass filters (Semrock TLP01-995). The magnetic field is created via an electro-magnet from GWM Associates (Model 5403EG-50) connected to a power supply from Danfysik (SYSTEM 8500 Magnet Power Supply). In this work a magnetic field of $B$ -0.21 T was used for measurements at sample B. If not indicated otherwise, we used 20 nW $A_2$ excitation power before the cryostation. The used detectors are fibre-coupled superconducting nanowire single-photon detectors from Photon Spot.

### Silicon carbide samples

Sample A had 10 μm epitaxial layer on 4H-SiC substrate and has natural abundance of silicon (4.7% $^{29}$Si) and carbon (1.1% $^{13}$C) isotopes, which are spin $I = 1/2$ nuclei. Samples were electron irradiated (5 kGy, 5 MeV) and annealed at 600 °C in 850 mbar Argon atmosphere. 1 μm width waveguides have been created by angled RIE. Sample B is almost identical besides of an implantation dose of 20 kGy. For the fabrication of the solid immersion lenses, a gallium FIB machine was used. Schottky contacts have been created by masked electron-beam physical vapour deposition of gold with a thin adhesion layer in between. The bottom ohmic contact is formed by the contact of silver paste with the highly doped SiC substrate.

## Data availability
The data sets that support the findings of the work presented in the article and its supplementary information are available from the corresponding author upon request.

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

## Acknowledgements
P.K., J.U.H. and J.W. acknowledge support from the European Commission through the QuantERA project InQuRe (Grant agreements No. 731473, and 101017733). T.S., J.U.H., V.V. and J.W. acknowledge support from the European Union's Horizon Europe research and innovation program through the SPINUS project (Grant agreement No. 101135699). P.K. and J.W. acknowledge the German ministry of education and research for the project InQuRe (BMBF, Grant agreement No. 16KIS1639K). J.W. acknowledge support from the European Commission for the Quantum Technology Flagship project QIA (Grant agreements No. 101080128, and 101102140), the German ministry of education and research for the project QR.X (BMBF, Grant agreement No. 16KISQ013) and Baden-Württemberg Stiftung for the project SPOC (Grant agreement No. QT-6). J.W. also acknowledges support from the project Spinning (BMBF, Grant agreement No. 13N16219) and the German Research Foundation (DFG, Grant agreement No. GRK2642). J.U.H. further acknowledges support from the Swedish Research Council under VR Grant No. 2020-05444 and Knut and Alice Wallenberg Foundation (Grant No. KAW 2018.0071). M.S. and M.B. acknowledge financial support from the Austrian Science Fund (FWF, grant I5195) and German Research Foundation (DFG, QuCoLiMa, SFB/TRR 306, Project No. 429529648).

## Author contributions
Methodology: T.S., P.K., E.H.-H., D.L., V.V., J.W. Measurements: T.S., P.K., E.H.-H., D.L., V.V. Fabrication: P.K., D.L., R.S. Sample preparation: T.S., P.K., D.L., R.S., W.K., M.G., J.U.-H. Writing, reviewing & editing: T.S., P.K., E.H.-H., V.V., J.W. Data analysis: T.S., P.K., V.V. Theoretical support & writing: M.S., M.B., G.B., A.G.

## Funding

## Competing interests
The authors declare no competing interests.
