## [Transparent Peer Review file · Nature Communications]

Single V2 defect in 4H Silicon Carbide Schottky diode at low temperature

Corresponding Author: Dr Vadim Vorobyov

Version 0:

Reviewer comments:

Reviewer #1

(Remarks to the Author)

The authors report optical and spin characteristics of V2 defects in 4H-SiC shottky diode. Stark shifts and PLE line narrowing were observed upon the application of the negative bias. The charge state of the defect was also investigated by addition of another laser. A solid immersion lens is integrated in a shottky diode, showing higher counts at a bias. The results are important for the development of solid state based quantum network nodes. However, defects in a SiC pin device and similar effects (Stark shift and line narrowing) have been already reported. An advantage of the present device structure is unclear compared with the reported pin diode, considering the obtained results. This point must be more clearly explained.

Additional comments.

- Please describe the schottky barrier between the metal contacts and SiC, and leak current under the applied negative bias up to -150 V.
- SIL in the device (Fig. 4(a)): The SIL structure should modifies the electric field distribution formed under the negative bias. Under -30 V used, how much electric field is generated at the defect position, compared with the flat device?

Reviewer #2

(Remarks to the Author)

The authors demonstrated Stark tuning and charge environment control of silicon vacancy centers in SiC using a diode structure. Moreover, the authors investigate the dynamics of the charge trap near the vacancy centers and explored the integration of the diode and the solid immersion lenses and the qubit manipulation, which is an important step towards quantum applications. I think the manuscript is suited for Nature Communications, and I only have a few minor comments:

1. The color code of the Figure 1 (d) is confusing, I would recommend a rainbow color code or something similar to obtain a clear trend of temperature variations.
2. The authors mentioned the heating problem with increased current in the diode. A quantitative power estimation versus the cooling power of the cryostat is preferred as it is an important factor regarding the tuning range and the emitter performance.
3. I'm not sure if the existence of Figure 1 (e) is necessary as the important details are shown in Figure 2 and SI. In fact, it is hard to say, by looking at the figure, to visually identify the single defects as the authors did. Otherwise, authors should elaborate on how to identify individual centers based on such a map.
4. The defect-2 used to assemble Figure 2a has detailed information in the SI, but in the main text, it was barely mentioned. A proper link should be given. In addition, it's hard to distinguish the dark blue dots with the regular blue dots. Another color should be used here to improve visibility.
5. In Figure 2 (c). What is the zero detuning point?
6. The caption of Figure 2 (d) looks confusing to me. It seems to be misaligned with the text.

7. What are the superscript numbers for the dynamical decoupling in Figure 4 (b) and Figure S4 (i)? The Figure S4 (i) fittings seem to have slightly different parameters.

Version 1:

Reviewer comments:

Reviewer #1

(Remarks to the Author)

I think the manuscript can be published.

Reviewer #2

(Remarks to the Author)

The revised manuscript is clearly improved from the previous version regarding the major claim of the manuscript and some of the figures, captions and legends. I also like the added analysis of the nature of the PLE switching which clearly indicates a nearby charge trap with varying electronic states.

I just want to clarify one point from my previous comment, which is Q7. for all echo and dynamical decoupling measurements, I am interested to learn more about the fitting parameters $\{n\}$ for the fitting curve, which I expect to be $\sim 1 - \exp\{-(T/T_2)^n\}$. As indicated by the fit curve pairs, this $\{n\}$ should be slightly different between the two traces (Are these from two centers? Or two pulse sequence logics for one center?). Some insights of the environment based on the measured $\{n\}$ is also preferred.

REVIEWER COMMENTS & ANSWERS

Reviewer #1 (Remarks to the Author):

The authors report optical and spin characteristics of V2 defects in 4H-SiC shottky diode. Stark shifts and PLE line narrowing were observed upon the application of the negative bias. The charge state of the defect was also investigated by addition of another laser. A solid immersion lens is integrated in a shottky diode, showing higher counts at a bias. The results are important for the development of solid state based quantum network nodes.

A: We are glad that the referee clearly understood the scope and the impact of our manuscript, in particular its importance for spin photon interfaces.

Q: However, defects in a SiC pin device and similar effects (Stark shift and line narrowing) have been already reported. An advantage of the present device structure is unclear compared with the reported pin diode, considering the obtained results. This point must be more clearly explained.

A: Indeed, the PIN diode structures were investigated previously with single VV (Di-Vacancy) centers and with ensembles of V1,V2 at room temperature, which we highlight in our literature review section. We think that the main contribution of our work is the combination of investigation of the charge state of the silicon vacancy under photo illumination and applied bias, its optical and spin properties. A more clear picture of the charge state dynamics and stability has been found out, which was only speculated before. Additionally, compared to previously reported work, the present device structure allows to reconstruct mentioned effects using a much simpler (in terms of manufacturing and scalability) fashion. It enables an easy fusion with required photonic technologies, which is an additional benefit of using a Schottky type of junction instead of P-I-N types.

To better underline this in the paper we have added the following paragraph to the discussion section:

“Our work gains insight into the charge state dynamics and stability of the silicon vacancy. This forms a more clear picture of the system, which was only speculated before. Additionally, the present device structure allows to reconstruct previously reported effects (e.g. Stark tuning, linewidth narrowing,...) using a much simpler manufacturing process. It allows chip-industry-like scalability and interoperability with required photonic technologies, which is an additional benefit of using a Schottky type of junction.”

Q: - Please describe the schottky barrier between the metal contacts and SiC, and leak current under the applied negative bias up to -150 V.

A: A metal-semiconductor junction realized by Gold on 4H-SiC forms a Schottky barrier with a height of 1.86eV using a work function of 5.1eV (Au) and electron affinity of 3.24eV (SiC). We have added a simplified (only considering the Epi layer) energy band structure of the junction at ambient and cryogenic conditions to the supplement (also see below). Already at

room temperature and zero bias we get a built-in voltage of 1.54 V causing a depletion area of $\sim 4.8\mu\text{m}$ depth. For -150 V bias the differences of metal and SC fermi levels rises drastically.. Typically, while we don't exceed the negative breakthrough voltage, which is the case for -150V if the diode didn't degrade, we don't have an observable reversed current. It is less than $0.1\mu\text{A}$ which is the limit of our ampere meter. Beyond that threshold voltage (approx. -250V) the current increases exponentially to several tens of mA, then our amplifier shuts down and is limiting the voltage/current.

Q: - SIL in the device (Fig. 4(a)): The SIL structure should modify the electric field distribution formed under the negative bias. Under -30 V used, how much electric field is generated at the defect position, compared with the flat device?

A: Yes, indeed, the presence of the SIL changes the distribution and the achieved electric field at the position of the defect. Moreover, the -30V limitation was in this case a technical limitation of the setup, but was sufficient to observe a line narrowing beyond the threshold of depletion. It is hard to compare the two geometries which we actually used in the experiment, as the defect depth, electric field orientation with respect to the c axis and contacts shape varied. In particular, the contact design was optimised for the best proximity to the defect, while not interfering with the optical structure. However, for simplified case, where everything is the same except the presence of the SIL, the simulation shows, that the electric field inside the SIL is atleast one order of magnitude less compared to location in bulk with the same distance to the electrode. In the experiment, we observe a similar behaviour, where the obtained line shifts are $\sim 1\text{-}2$ GHz in both devices. The reason why we don't see less shifts in the SIL is explained by the much closer electrodes and the electric field is not only orientated perpendicular to the c-axis.

Reviewer #2 (Remarks to the Author):

The authors demonstrated Stark tuning and charge environment control of silicon vacancy centers in SiC using a diode structure. Moreover, the authors investigate the dynamics of the charge trap near the vacancy centers and explored the integration of the diode and the solid immersion lenses and the qubit manipulation, which is an important step towards quantum applications. I think the manuscript is suited for Nature Communications, and I only have a few minor comments:

We are grateful to the referee for their accurate perception of our manuscript.

Q: 1. The color code of the Figure 1 (d) is confusing, I would recommend a rainbow color code or something similar to obtain a clear trend of temperature variations.

A: The standard color sequence of subsequent plots in python has been used, causing this irregular color code. Now we changed to a “rainbow color sequence” and connected data points to depict temperature dependence more clearly. The figure has been updated.

Q: 2. The authors mentioned the heating problem with increased current in the diode. A quantitative power estimation versus the cooling power of the cryostat is preferred as it is an important factor regarding the tuning range and the emitter performance.

A: According to UI curves, at 8-10K, the dissipated power in forward direction reaches ($P=U*I$) approx. 100 mW of electrical power, dissipating into the sample. With a silver paste of $k=100\text{W/m/K}$, sample size $A = 4\times 5 \text{ mm}^2$ and glue thickness $d = 10\mu\text{m}$, we expect excellent heat contact to the cryo cold plate, with $k*A/d = 200 \text{ W/K}$ heat conductance. However, our cryostat cooling power at 4.5 K is by specification sheet is around 100 mW of cooling power (<https://www.montanainstruments.com/products/h05>).

Possibly, at 8K, the cooling power is slightly better, however, we also expect the degradation of our cooling power, as the cold head resource is already at about 70%, hence it could produce less cooling capacity. Nevertheless, we note that heating is not happening, when the negative voltage is applied (for the bulk sample down to -250V possible), due to an extremely low leakage current ($P(@-250\text{V}) \sim < 0.025\text{mW}$). In that case, we expect the tuning capability of 3-5 GHz. Ultimately, the tuning range could be improved (by approx. 10x), when the - field is aligned with the crystal c-axis as was shown in previous work by Lukin et al.

Q: 3. I'm not sure if the existence of Figure 1 (e) is necessary as the important details are shown in Figure 2 and SI. In fact, it is hard to say, by looking at the figure, to visually identify the single defects as the authors did. Otherwise, authors should elaborate on how to identify individual centers based on such a map.

A: We agree that it was not really obvious to identify single defects from the confocal scan we show in Fig1 (e). As the figure also shows well the orientation of contacts and c-axis we decided to keep the figure, but in the revised manuscript we have added an inset showing the confocal micrograph a small area (dashed white rectangular) consisting of a few defects, showing that indeed each defect can be addressed individually optically. Furthermore, by resonant excitation, the defects can also be separated spectrally.

Q: 4. The defect-2 used to assemble Figure 2a has detailed information in the SI, but in the main text, it was barely mentioned. A proper link should be given. In addition, it's hard to distinguish the dark blue dots with the regular blue dots. Another color should be used here to improve visibility.

A: Indeed, e.g. the information about the distance to the electrodes of defect-2 were missing in the main text. The defect was located $\sim 4\mu\text{m}$ away from the electrode, but is not located inside the confocal we use in the main draft. Its position is now marked in the SM and linked in the main text. We modified the corresponding sentence in the result section:

“The extracted PLE position and linewidth is shown in Fig. 2a for two defects with defect-1 located $\sim 11.2 \mu\text{m}$ from the contact (red circle in Fig. 2b) and defect-2 with a distance of $\sim 4 \mu\text{m}$ to the electrode (see SM Fig. S1b).”

For a better visibility, we changed the color and marker in Fig. 2a. Now both data sets for the linewidths of both defects can clearly be distinguished.

Q: 5. In Figure 2 (c). What is the zero detuning point?

A: We show here the detuning of the resonant laser from some reference point (now 327.116 THz, measured with a wavemeter), called zero detuning point. For clarity, we have now subtracted the final position from the point of 0 bias volt and replotted the figure.

Q: 6. The caption of Figure 2 (d) looks confusing to me. It seems to be misaligned with the text

A: Indeed the caption is not as detailed as required to entirely understand our study. We investigate a single VSi center in the vicinity of at least one two-level system (TLS), in this case associated with a charge trap. By fast spontaneous switching of its state, the interaction with the color center leads to two stable PLE positions, which is shown in the inset of Fig.2(d). By laser illumination of the defect (and its vicinity) we observed a photo-induced reconfiguration of the TLS steady state. By this, The occupation probability of both states can be modified, and identified by the intensity of the corresponding PLE peak. We did a time-resolved study of that TLS by tracking one PLE intensity versus the duration of different applied lasers. We observe that on a millisecond time scale we depopulate the TLS into the opposite level, while we observe a recovery if no laser is applied.

We changed the caption to:

“Inset: PLE of a V2 center affected by a fast-switching two-level system (TLS) in its vicinity, again associated with a charge trap. Mainfigure: Time-resolved study of that TLS by tracking one PLE intensity versus the duration of laser illumination (red: no excitation; orange: 20 μW off-resonant excitation; blue: 3 nW of resonant to V2 excitation). Depopulation of the corresponding TLS state (decrease in PLE intensity) by photo-induced reconfiguration, while recovery if no laser is applied.”

Q: 7. What are the superscript numbers for the dynamical decoupling in Figure 4 (b) and Figure S4 (i)? The Figure S4 (i) fittings seem to have slightly different parameters.

A: As we could not precisely identify which superscripts were asked by the reviewer, we concluded that it must be the x-axis tick labels. In figure 4(b) and S4(i) we have plotted the coherence of spin versus the interpulse time τ , on a logarithmic scale of time, with a base of 10. Another superscript appearing in the Figure S4 (i):

$N^{2/3}$ used as a scaling, like in the previous work (G. de Lange et. al., Science 330, 60 (2010)) and (0V, -30V) referring to applied bias (xlabel of plot has been adjusted). Indeed the fittings do show slightly different parameters, but 1) they are not high enough for making a claim of better coherence properties, 2) in our samples the coherence probably is limited by the nuclear spin bath and not electronic noise, 3) the coherence properties did not become worse, rather slightly even improved. (see figure S4 (j).)

Elaborated charge discussion:

During the reviewing-process we finished supporting calculations which elucidate our charge trap study in Fig. 2c) by the presence of a carbon vacancy in a distance of approx 50 nm. Therefore we have added a figure to the supplement and adjusted the discussion section: "We attribute the beneficial charge robustness of the silicon vacancy to its favourable environment. The available charge traps, namely carbon vacancy, in its vicinity support the VSi to be most of the time in the correct charge state. Investigations of the probability of ionisation of the carbon vacancy (see Fig. S7) reveals the defect to be a donor-like neighbour under photo illumination, further exclude VSi⁰. The charge transition behaviour between these three state, fits to the observed results of PLE distribution (see Fig. 2c) for a V2 affected by the emerging electric field discrepancies."

As the calculations have been done mainly by Guodong Bain, we added his name to the author list.

This result completes the charge discussion and does not change the focus nor the statement of the paper. By that, we hope to get the consent of the reviewer for the changes we made.

List of all changes done in the manuscript:

1. Changes in the main text and captions are highlighted
2. Larger text changes has been made for:
 - a. Caption of Fig2d - due to poor description, identified by the reviewer
 - b. Discussion section - to emphasize the remarks of the paper like suggested & adding paragraph to new supplement data
3. Colorcode has been changed in Figure 1d
4. Inset has been added to Figure 1e
5. Color and style of markers has been changed in Figure 2a
6. x offset has been changed in Figure 2c
7. Guodong Bian has been added to the author list
8. Minor text changes has been applied to supplement (marked)
9. Chapter V. and Figure S7 has been added to the supplement
10. Band structure plots have been added to Fig. S1

REVIEWER COMMENTS & ANSWERS #2

Reviewer #1 (Remarks to the Author):

Q: I think the manuscript can be published.

A: We are grateful to hear that the reviewer is fully satisfied with our manuscript

Reviewer #2 (Remarks to the Author):

Q: The revised manuscript is clearly improved from the previous version regarding the major claim of the manuscript and some of the figures, captions and legends. I also like the added analysis of the nature of the PLE switching which clearly indicates a nearby charge trap with varying electronic states.

A: We thank the reviewer for the nice feedback and the acceptance of our added charge trap study.

Q: I just want to clarify one point from my previous comment, which is Q7. for all echo and dynamical decoupling measurements, I am interested to learn more about the fitting parameters $\{n\}$ for the fitting curve, which I expect to be $\sim 1 - \exp\{-(T/T_2)^n\}$. As indicated by the fit curve pairs, this $\{n\}$ should be slightly different between the two traces (Are these from two centers? Or two pulse sequence logics for one center?). Some insights of the environment based on the measured $\{n\}$ is also preferred.

A: Regarding the pairs of data sets/fits in figure 4b, those are all on the same defect, where each pair belongs to a DD sequence measured for different bias voltages (0V/-30V). Just from this plot it is not explicit that the coherence times improve a little, as well as changes in the exponential decay index n (fit $\sim \exp\{-\{\tau/T_{\text{coh}}\}^n\}$) can not be reliably identified. We added a plot below, showing the exponent $\{n\}$ against the number of DD pulses for both bias voltages (the pairs of 4b corresponds then to 2 data points for the same pulse number).

The trend line shows that for increasing pulse numbers the exponent increases from the Gaussian envelope regime $n=2$ to $\sim n=3$.

It is a common knowledge, that for faster electronic noise, in general the decay stretch exponent of power $n=3$ is expected [1] while for slow diluted nuclear spin bath, as presented here, a power of $n = 3-4$ is expected [2], as despite having 4% natural abundance, the relative distance between Si is 2-3 times larger, and the respected gyromagnetic factor is 0.8 times weaker, hence the intra nuclear interaction is analogous to 1% of ^{13}C bath in diamond lattice. We suggest that as the uncertainty of the extracted power factor n is from 2 to 4 for some points, and the values are oscillating, especially, between HE, XY16 and further to longer sequences, while it is appealing to speculate regarding the change of the environment, in reality, it is not very convincing, and our study more shows that the coherence properties of the obtained qubit register created by electron and nuclear spin are preserved.

[1]: <https://www.science.org/doi/10.1126/science.1192739>

[2]: <https://journals.aps.org/prb/pdf/10.1103/PhysRevB.78.094303>